# Effect of the Abdominal Draw-In Maneuver and Bracing on Abdominal Muscle Thickness and the Associated Subjective Difficulty in Healthy Individuals

**DOI:** 10.3390/healthcare8040496

**Published:** 2020-11-19

**Authors:** Sachiko Madokoro, Masami Yokogawa, Hiroichi Miaki

**Affiliations:** Division of Health Sciences, Graduate School of Medical Science, Kanazawa University, Kanazawa 920-0942, Japan; yokogawa@mhs.mp.kanazawa-u.ac.jp (M.Y.); miaki@mhs.mp.kanazawa-u.ac.jp (H.M.)

**Keywords:** draw-in maneuver, bracing, transversus abdominis, subjective difficulty, stability exercise, posture

## Abstract

We evaluated the trunk abdominal muscle thickness while performing different exercises to identify the most effective training and to investigate the subjective difficulty associated with exercising. Twenty-eight men (mean age: 21.6 ± 0.9 years) without orthopedic diseases were enrolled. Ultrasonic imaging was used to measure the thickness of the transversus abdominis (TA), internal oblique, and external oblique muscles while at rest and while performing the abdominal draw-in maneuver and abdominal bracing. Measurements were made in the supine and sitting positions, and the subjective difficulty in performing each exercise was examined using a 5-level evaluation scale. The TA and internal oblique muscle thicknesses were significantly greater during the abdominal draw-in maneuver (ADIM) than during bracing or resting, in the supine and sitting positions. The subjective difficulty of abdominal bracing (AB) was graded significantly higher than that of ADIM. Additionally, a correlation between subjective difficulty and muscle thickness was found for the TA and IO. Our results may contribute to the choice of more effective exercises for spinal stability.

## 1. Introduction

Spinal stabilization exercises improve spinal stability by improving trunk muscle coordination and anchoring the deep musculature, which prevents overactivity by the superficial musculature [1,2]. Meta-analyses have shown that spinal stabilization exercises are among the most effective forms of physiotherapy for improving lower back pain [3,4]. The deep musculature consists of the transversus abdominis (TA), internal oblique (IO) muscles, multifidus, diaphragm, and pelvic floor muscles, which comprise the so-called “inner unit” [5]. 

In recent years, various spinal stabilization exercises have been proposed. Abdominal draw-in maneuver (ADIM) is one such exercise that increases the activity of these muscles. Many previous studies have shown that it is effective for alleviating back pain [6,7,8]. Gorbet et al. [9] divided their participants into asymptomatic low back pain patients and healthy individuals using the Oswestry Disability Index score and low back pain-free period. They reported that in asymptomatic patients with low back pain, the action of TA was similar between the two groups, but ADIM enhanced the action of the TA in both groups. Kato et al. [10] reported the effects of wrapping a cuff around the abdomen and resisting it to inflate the abdomen as a new core exercise. Immediately after performing this abdominal expanding exercise, positron emission tomography-computer tomography (PET-CT) showed significant activity of the rectus abdominis and diaphragm. Especially, they indicated this exercise as a stabilization exercise for abdominal bracing. Other studies have also reported that abdominal bracing by contracting all the abdominal muscles simultaneously could improve the trunk stability. Grenier et al. [11] used electromyography and movement simulation to show that abdominal bracing (AB) improves trunk stability in comparison with ADIM. Other studies have also shown that AB is more effective in increasing intra-abdominal pressure, which is regarded as important for trunk stability [12]. These spinal stabilization exercises have been shown to be effective in increasing core muscle activity, but their relationship with subjective difficulties remains unclear. Therefore, teaching spinal stabilization exercises in clinical practice may be difficult. Good adherence is necessary to improve the effectiveness of exercise [13]. Saner et al. [14] knowledge is required to perform exercises correctly and ensure home exercise adherence by these patients. Beinart et al. [15] stated that adherence to exercise therapy was low, at 50–70%. We believe that the difficulty of mastering an exercise regimen is a probable explanation to the observed low adherence percentage. Lack of confidence in performing the exercise correctly was previously described as a key barrier to successful compliance [16,17,18,19,20]. Therefore, easy-to-learn exercises could be more easily performed at home and may improve adherence. 

Therefore, the purpose of this study was to clarify the relationship between the subjective difficulty of performing ADIM and AB exercises and the lateral abdominal muscle thickness during exercise. In this study, the abdominal expanding exercise was regarded as one of the bracing exercises with reference to the method by Kato et al. In some previous studies, diagnostic ultrasound equipment was used to measure muscle thickness. McMeeken et al. [21] stated that muscle thickness was strongly correlated with muscle activity (R^2^ = 0.87). Studies have confirmed the correlation between abdominal muscle thickness and muscle activity [22,23,24]. At the same time, we also investigated the postures that make it more difficult to stabilize the spinal column. The Likert 5-point scale method is used for questionnaire survey methods in various fields. Regarding the subjective difficulty, there are previous studies by Dooley et al. [25], Sakai et al. [26], and others. Nagata et al. [27] reported that the 5-point method is more reliable than the 4-point and 7-point methods. Therefore, in this study, we decided to measure the subjective difficulty of the participants’ movement using the 5-point method. Our hypothesis was that the subjective difficulty of stabilization exercises would vary from person to person. Moreover, we hypothesized that exercises that the patients would find difficult to perform would not sufficiently thicken the patients flank muscles.

## 2. Materials and Methods

### 2.1. Study Design

This is a cross-sectional study comparing the effects of ADIM and AB on young, healthy men.

### 2.2. Study Participants

The study participants were 28 adult men (mean age, 21.6 ± 0.9 years; mean height, 174.0 ± 7.5 cm; mean weight, 64.7 ± 6.2 kg; mean waist circumference, 74.6 ± 6.2 cm) with no orthopedic disease at the time of the study. Individuals with a history of trunk surgery or neurological disease were excluded. 

We calculated the sample size needed for the two-way analysis of variance (ANOVA) with repeated measures (effect size, 0.25; α error, 0.05; power, 0.8) using the G*power software (Version 3.1., Heinirich Heine University, Duesseldorf, Germany). The results showed that 28 participants were required.

The study was conducted after all participants received an explanation in writing. The study was conducted in accordance with the Declaration of Helsinki as revised in 2013. Verbal and written informed consents were obtained from all participants. This study was approved by the Kanazawa University Medical Ethics Committee (approval number: 862-1; date of approval: 10 August 2018).

### 2.3. Measurement Procedure

All measurements were made in a laboratory in the Kanazawa University. A MyLab25 digital diagnostic ultrasound device (Esaote, Indianapolis, IN, USA) was used to measure trunk lateral abdominal muscle thickness. Measurements were carried out with the participants in two different postures: (1) supine position, with the knees at an angle of 90° flexion; and (2) sitting position, with both hips and knees at an angle of 90° flexion. Following previous ultrasound studies, muscle thickness was measured on the right side of the abdomen, at a point marked on the skin on the right anterior axillary line at the midpoint of the rib margin and the iliac crest. The position and angle of the probe were adjusted so that the boundaries of the abdominal muscles were clearly visible, and the muscle thickness was visualized horizontally as far as possible. Scanning was performed for 10 s. Images scanned at the end of expiration were extracted, and muscle thickness was measured to the nearest 0.1 mm using the ImageJ software (NIH, Bethesda, MD, USA) (Figure 1). The midpoint of each image was selected to ensure that the same measurement location was used for individual participants [28,29,30]. All ultrasound measurements were performed by a single examiner who was a physiotherapist. A different examiner conducted ultrasound measurement analysis.

Measurements were made at rest and while performing ADIM and AB exercises. Following the study by Tayashiki et al. [12], the participants were instructed to draw in their umbilicus toward their spine when breathing out, and ultrasound was used to check that they had properly understood this procedure before measurements were performed during ADIM. ADIM was practiced until the echo confirmed that the TA and IO muscle thicknesses had increased. The number of exercises performed depended on each participant’s capability. Following the study by Kato et al. [10], before measurements during AB exercise, the participants were instructed to inflate their abdomen and tighten their core muscles while breathing out. Measurements were made after confirmation that this procedure was properly understood. The participants practiced inflation of the abdomen so that the examiner lightly pressed the flank and pushed back the examiner’s hand. In this study, the strength of AB was set to a level where motion could be maintained for 10 s. Measurements were made in a random order in the supine and sitting positions (Figure 2).

After all exercises had been completed, the participants were asked to rate the difficulty of each exercise on a 5-point scale (very easy, quite easy, routine, quite difficult, or very difficult) and to comment in their own words.

### 2.4. Statistical Analysis

Statistical analyses were performed to compare the thicknesses of the TA, IO, and external oblique (EO) muscles, while resting and during ADIM and AB exercises in the supine and sitting positions. To confirm the reliability of measurements, the intraclass correlation coefficients (ICCs) for each position and exercise were first calculated. Then, two-way ANOVA was then conducted for each muscle to check for interactions. In cases where two-way ANOVA revealed a main effect, one-way ANOVA between each of the exercise conditions was performed, with Bonferroni’s multiple comparisons being used as a subtest. When the position was found to have a main effect, a paired t-test was used for comparisons between the supine and sitting positions. The difficulty of performing each exercise was compared using one-way ANOVA for comparisons between ADIM and AB exercises in each of the supine and sitting positions, with Bonferroni’s multiple comparisons being used to make the subsequent tests. Spearman’s correlation coefficient was calculated for the subjective difficulty and muscle thickness. Moreover, the partial correlation coefficient for the correlation between the subjective difficulty level and the muscle thickness under each condition (ADIM, AB, supine position, sitting position) was obtained. SPSS version 23 (IBM Corp., Armonk, NY, USA) was used for all statistical analyses, and the level of significance was set at *p* < 0.05. There were no missing data in this study.

## 3. Results

### 3.1. Subjective Difficulty of Exercises 

In both the supine and sittign positions, AB exercise was rated as significantly more difficult compared to ADIM (Table 1).

### 3.2. Intraclass Correlation Coefficients

Table 2 shows the ICCs and 95% confidence intervals (95% CIs) for lateral abdominal muscle thickness measurements. The ICC was ≥0.85 for all measurements, indicating high reliability (Table 2).

### 3.3. Lateral Abdominal Muscle Thickness

Two-way ANOVA did not reveal any interactions between the measured thicknesses of the TA, IO, and EO (TA: F = 0.374, *p* = 0.689; IO: F = 1.954, *p* = 0.145; EO: F = 1.366, *p* = 0.258). The main effect was evident between the TA and IO under all exercise conditions. Table 3 shows the different exercise conditions in the supine and sitting positions. In the supine and sitting positions, the TA and IO were significantly thicker during ADIM than at rest. Moreover, they were significantly thicker during ADIM than during AB exercise. There was no significant difference in their thickness while being at rest and during AB exercise. The EO was significantly thicker during ADIM than during AB in the supine position. There was a main effect of positioning on IO thickness. A comparison between the supine and sitting positions showed that the IO was significantly thicker in the sitting than in the supine position during ADIM and AB exercises. The EO was significantly thicker in the sitting than in the supine position during AB exercise. There was no difference in the TA thickness between the two positions (Table 4).

### 3.4. Correlation of Subjective Difficulty and Muscle Thickness

Regarding muscle thickness data, including all conditions and subjective difficulty, a significant correlation was observed between the IO and the TA and the corresponding subjective difficulties. Regarding the muscle thickness data and the subjective difficulty in each condition, a significant correlation was found only between the TA muscle thickness and subjective difficulty in the supine position (Table 5).

## 4. Discussion

We found that the subjective difficulty of AB exercise was graded significantly higher compared to ADIM. To date, various stability exercises have been reported. However, to the best of our knowledge, this is the first report focused on the subjective difficulty level. We predicted that the subjective difficulty of stability exercises would vary widely, but the results of this study showed that AB exercise was more difficult. Our study participants were healthy adult men who had no difficulties in learning the exercises. However, most of the participants felt that AB was more difficult. Many of those who responded that AB exercise had increased difficulty explained that the performance of this exercise involved a breathing technique that they did not perform in everyday life. Exercise adherence is important for patients with lower back pain [15]. Indeed, according to Saner et al. [14] knowledge is required to perform exercises correctly and ensure home exercise adherence by these patients. Patients who have difficulties in performing ADIM may face various situations, but one option is to use visual feedback using ultrasound. However, our results suggested that it may be difficult to provide ultrasound feedback for AB. This suggested that guidance may be necessary to carry out bracing exercises effectively. Its difficulty may also increase with age.

In this study, we investigated the changes in the thickness of the TA, IO, and EO during ADIM and AB exercises in the supine and sitting positions. In both these positions, the thicknesses of the TA and IO increased significantly during ADIM. This result was consistent with those of previous studies [23,28] and was attributed to the increased muscle activity of the TA and IO. However, there was no significant increase in the thickness of the TA, IO, or EO during AB exercise compared with the thickness while resting. Aboufazeli et al. [31] used ultrasound to investigate the reproducibility of muscle thickness measurements during ADIM and AB exercises and reported that both were reproducible. As in our study, they also found that muscle thickness was greater during ADIM than during AB exercise. Koh et al. [32] used CT to compare ADIM and AB exercise and found that the TA area increased more during ADIM, whereas bracing had a greater effect on the rectus abdominis, IO and EO. Kato et al. [10] used PET-CT to investigate uptake immediately after performing AB exercise and found that the rectus abdominis and the diaphragm engaged in significant work. Therefore, our results suggested that AB had greater effect on other muscles, such as the rectus abdominis muscle and the diaphragm, rather than the TA, IO and EO.

Using surface electromyography, Tayashiki et al. [33] and Maeo et al. [34] reported that the rectus abdominis, EO, IO, and erector spinae muscles were working during AB. Variances from our results may have been observed due to differences in the given verbal instructions. Participants were instructed to expand and brace the abdomen or brace the abdomen as a whole without sucking it in. Therefore, further investigations into the verbal instructions given for AB should be conducted. AB was associated with increased intra-abdominal pressure [12]. Elevated intra-abdominal pressure improved spinal stability and contributed to postural stability [35,36]. Kim et al. [37] found that both AB and ADIM contributed to spinal stability when they were performed during a sudden shift in the supporting surface. Kim et al. [38] reported that performing both ADIM and AB during stability exercises was effective for older women with nonspecific lower back pain. Thus, ADIM and AB may benefit spinal stability, as intra-abdominal pressure and spinal stability were not measured in this study. This is a topic for further investigation. In the future, it is necessary to study the stability of the spinal column. When intra-abdominal pressure increased during AB, the pelvic floor muscles also reportedly dropped [39,40]. Thus, this maneuver may not be indicated in patients with stress urinary incontinence. 

A comparison of the different positions revealed that the IO was significantly thicker while the individual was sitting during both ADIM and AB exercises. In contrast, the EO was significantly thicker during AB only in the sitting position. There was no significant difference between the thickness of the TA in the supine and sitting positions while performing any exercise. Maeo et al. [37] investigated muscle activity during ADIM, AB, and other exercises performed in the standing position and found that both ADIM and AB produced high muscle activity in the IO. This finding showed that high muscle activity in the IO may be produced during exercise in an anti-gravitational posture. The TA tended to be thicker in the sitting than in the supine position at rest, although this difference was not significant (*p* = 0.08). Reeve et al. [41] reported that the thickness of the TA increased more in the sitting position than in the supine position using ultrasound. However, our results showed no difference. As the IO was working significantly harder during exercises in the sitting position, there may have been no relative changes in the TA thickness. Further studies on changes in the trunk musculature as a whole are required.

Additionally, a correlation between subjective difficulty and muscle thickness was found for the TA and IO. These results were the first to suggest that the contraction of deep core muscles is related to the ease of implementation. Conversely, there was no significant correlation between muscle thickness and subjective difficulty under each condition. For ADIM, a relatively high correlation coefficient was observed for TA contraction in the supine position compared to the sitting position (*p* = 0.05), suggesting that subjective difficulty may have an effect. Conversely, no correlation was found between the TA muscle thickness during ADIM in the sitting position and the subjective difficulty level during ADIM. Previous studies have reported that the upright sitting position increases muscle activity of the trunk muscles compared to the supine position [42,43]. Muscle activity to maintain posture may result in increased muscle thickness regardless of whether the participant experiences difficulty in exercising. These results suggested that it is more important for the participant to properly understand and perform the exercise in the supine position. When performing the AB exercise, the correlation coefficient between subjective difficulty and muscle thickness of IO and TA was low. It was suggested that the AB exercise performed in this study did not affect muscle thickness regardless of the subjective difficulty level. Similarly, our results suggested that it is necessary to feel that ADIM can be easily performed in the supine position to obtain intensive muscle contraction of the deep core muscles. Further studies on exercise instruction methods may be required.

The limitations of this study included the fact that all our participants were young, healthy men, and the sample size was small. Moreover, only three muscles (the TA, IO, and EO muscles) were measured. Therefore, we were thus unable to investigate sex differences, the effect of training in older people, or the activity of other muscles. In addition, we performed the exercise to inflate the abdomen as AB with reference to the method of Kato et al. However, we did not investigate the subjective difficulty level when changing the method, such as wrapping something around the abdomen. We have also not investigated the way in which the differences in the provided verbal instructions could affect the subjective difficulty. Thus, we should analyze whether the subjective difficulty of AB is related to the use of various verbal cues. Finally, the difficulty was graded subjectively, without the use of salivary amylase or other biochemical analyses. Further studies are required in light of these limitations.

## 5. Conclusions

The purpose of this study was to clarify the relationship between the subjective difficulty of stability exercise and flank muscle thickness. The AB exercises were more subjectively difficult compared to ADIM exercises. Additionally, a correlation between the subjective difficulty and muscle thickness was found for the TA and IO. In rehabilitation, the most effective method must be chosen for the patient concerning the purpose. Based on the results and those of previous studies, ADIM is considered an inner muscle training exercise that includes the pelvic floor muscles and is relatively easy to perform. It is not clear whether ADIM and AB can be easily performed by patients with low back pain. However, even in young, healthy people, there are individual differences in the ease of performing these exercises, which may affect the exercise effect. Our results may contribute to suggest more effective exercises for cases of spinal stability. These findings should be considered when devising exercise regimes for subjects with lower back pain and various other conditions.

## Figures and Tables

**Figure 1 healthcare-08-00496-f001:**
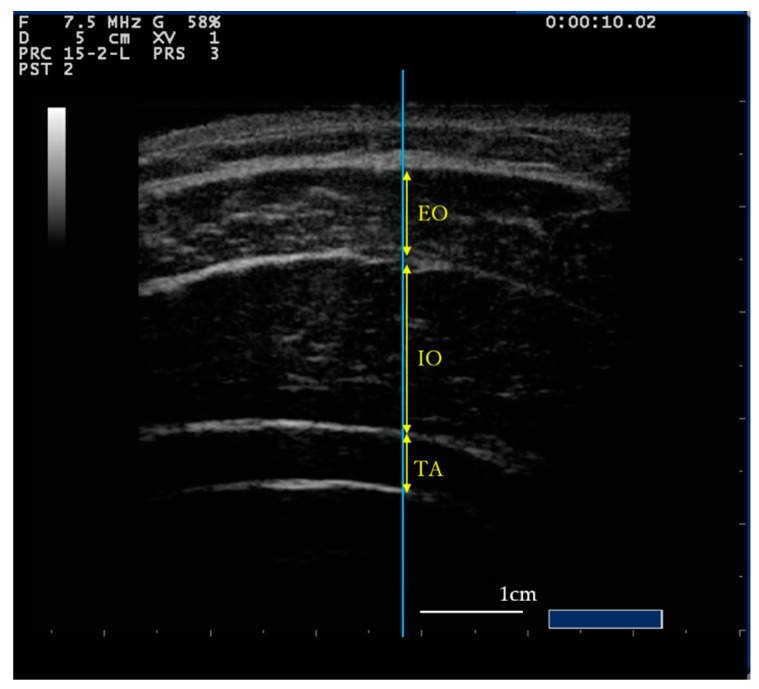
Ultrasound image of the flank. The Blue line passing through the middle of the image. The yellow arrow indicates the measurement site of each muscle. The white line shows the 1-cm scale bar. During the measurement, the position of the probe was kept constant. EO, External oblique muscle; IO, Internal oblique muscle; TA, Transverse abdominal muscle.

**Figure 2 healthcare-08-00496-f002:**
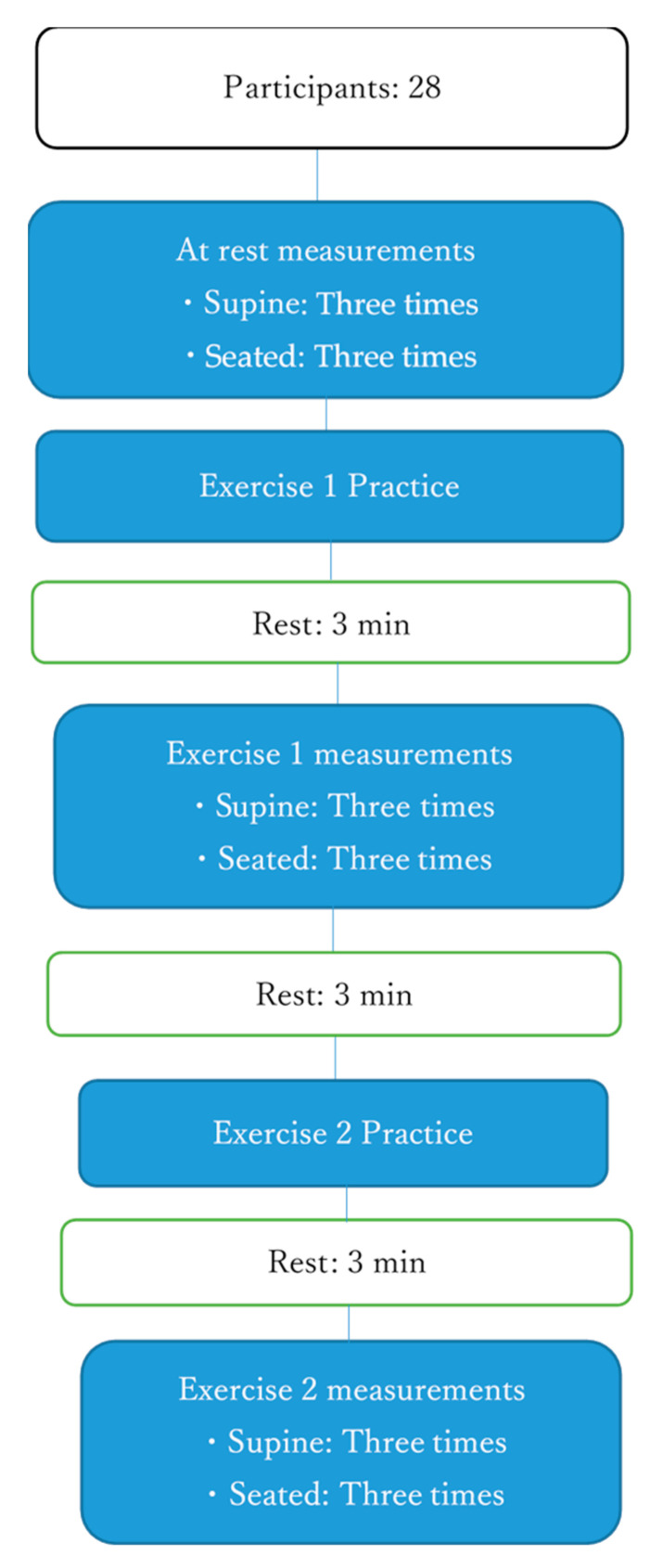
Experimental flow. The participants were randomly assigned to perform Exercise 1 and Exercise 2 to determine the difficulty of performing abdominal draw-in maneuver (ADIM) and abdominal bracing (AB) exercises. The measurements for the sitting or supine position were randomly selected.

**Table 1 healthcare-08-00496-t001:** Subjective difficulty.

Posture	Supine		Sitting	
	ADIM	AB	ADIM	AB
Difficulty	2.0 ± 1.0	3.26 ± 0.9 *^,†^	2.15 ± 0.91	3.19 ± 0.83 *^,†^

The values are presented as means ± standard deviations. ADIM: abdominal draw-in maneuver; AB: abdominal bracing. One-way ANOVA between the different exercise conditions in each position, followed by Bonferroni’s subtest. * Comparison with supine ADIM *p* < 0.05. ^†^ Comparison with sitting ADIM *p* < 0.05.

**Table 2 healthcare-08-00496-t002:** Intraclass correlation coefficient (95% CI).

Posture	Supine			Sitting		
Muscle	Rest	ADIM	AB	Rest	ADIM	AB
TA	0.95 (0.90–0.98)	0.95 (0.91–0.98)	0.96 (0.93–0.98)	0.94 (0.89–0.97)	0.96 (0.93–0.98)	0.87 (0.75–0.93)
IO	0.98 (0.96–0.99)	0.91 (0.84–0.96)	0.94 (0.89–0.97)	0.97 (0.94–0.98)	0.97 (0.95–0.98)	0.96 (0.93–0.98)
EO	0.97 (0.95–0.99)	0.88 (0.78–0.94)	0.95 (0.91–0.98)	0.95 (0.91–0.98)	0.90 (0.80–0.95)	0.92 (0.85–0.96)

Intraclass correlation coefficients for the same participants in the same day. CI: confidence interval; ADIM: abdominal draw-in maneuver; AB: abdominal bracing; TA: transversus abdominis; IO: internal oblique; EO: external oblique.

**Table 3 healthcare-08-00496-t003:** Thickness of the abdominal muscles (mm).

Posture	Supine					Sitting				
	Rest	ADIM	AB	Comparison	*p*	Rest	ADIM	AB	Comparison	*p*
TA	4.0 ± 0.8	6.9 ± 1.7	4.9 ± 2.0	Rest-ADIM	<0.01	4.3 ± 1.3	6.7 ± 2.4	4.7 ± 1.8	Rest-ADIM	<0.01
				Rest-AB	0.14				Rest-AB	1.00
				ADIM-AB	<0.01				ADIM-AB	<0.01
IO	10.1 ± 1.9	12.1 ± 2.5	10.5 ± 2.6	Rest-ADIM	<0.01	10.9 ± 3.0	15.0 ± 3.6	11.8 ± 4.2	Rest-ADIM	<0.01
				Rest-AB	1.00				Rest-AB	0.96
				ADIM-AB	0.03				ADIM-AB	<0.01
EO	7.0 ± 1.3	7.8 ± 1.4	6.9 ± 1.6	Rest-ADIM	0.12	7.3 ± 1.4	7.5 ± 1.2	7.4 ± 1.4	Rest-ADIM	1.00
				Rest-AB	1.00				Rest-AB	1.00
				ADIM-AB	0.04				ADIM-AB	1.00

The values are presented as means ± standard deviations. One-way ANOVA between the different exercise conditions in each position was performed, followed by Bonferroni’s subtest. ADIM, abdominal draw-in maneuver; AB, abdominal bracing; TA, transversus abdominis; IO, internal oblique; EO, external oblique.

**Table 4 healthcare-08-00496-t004:** Muscle thickness (between positions).

Muscle	Exercise	Supine	Sitting	*p*
TA	Rest	4.0 ± 0.8	4.3 ± 1.3	0.08
	ADIM	6.9 ± 1.7	6.7 ± 2.4	0.78
	AB	4.9 ± 2.0	4.7 ± 1.8	0.52
IO	Rest	10.1 ± 1.9	10.9 ± 3.0	0.06
	ADIM	12.1 ± 2.5	15.0 ± 3.6	<0.01
	AB	10.5 ± 2.6	11.8 ± 4.2	0.02
EO	Rest	7.0 ± 1.3	7.3 ± 1.4	0.13
	ADIM	7.8 ± 1.4	7.5 ± 1.2	0.19
	AB	6.9 ± 1.6	7.4 ± 1.4	0.02

Comparisons of muscle thicknesses during the different exercises in each position using a t-test. TA, transversus abdominis; ADIM, abdominal draw-in maneuver; AB, abdominal bracing; IO, internal oblique; EO, external oblique.

**Table 5 healthcare-08-00496-t005:** Correlation coefficient between muscle thickness and subjective difficulty.

Posture		Supine		Sitting	
	All (*p*)	ADIM (*p*)	AB (*p*)	ADIM (*p*)	AB (*p*)
TA	−0.400 (<0.01)	−0.373 (0.05)	−0.223 (0.25)	−0.051 (0.80)	−0.275 (0.16)
IO	−0.284 (<0.01)	−0.054 (0.79)	−0.050 (0.80)	−0.078 (0.69)	−0.313 (0.11)
EO	−0.110 (0.25)	0.39 (0.84)	−0.050 (0.80)	−0.51 (0.80)	−0.126 (0.52)

Spearman’s correlation coefficient results for the correlation between each muscle thickness data and the subjective difficulty in all conditions are presented. Regarding the correlation between each muscle thickness and the subjective difficulty, Pearson’s correlation and Spearman’s correlation coefficients for normal and non-normal distribution data were used, respectively. ADIM, abdominal draw-in maneuver; AB, abdominal bracing; TA, transversus abdominis; IO, internal oblique; EO, external oblique.

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
