# Peer review of "Effect of the Abdominal Draw-In Maneuver and Bracing on Abdominal Muscle Thickness and the Associated Subjective Difficulty in Healthy Individuals"

_healthcare, 2020, doi:10.3390/healthcare8040496_

Round 1

Reviewer 1 Report

Introduction

The state of the art in the area studied must be improved, explaining actual knowledge and the necessity of the present research

The objective must be concise, according to the introduction proposed

Include a study hypothesis

Methods

Include ± in height values

ANOVA no ANNOVA

For better comprehension procedure must be in deep described and I would recommend including a figure to visually explain the procedure

Results

Place the p exact value in tables where p < 0.05 is used

Discussion

Explain if the hypothesis is confirmed and why

Discuss results obtained and explain differences according to previous studies

Conclusion

Must be concise, responding to the study aims

Author Response

Dear Reviewer 1

I am very sorry for the late submission. I have attached the fix file. 

Please see the attachment. Thanks you very much.

Reviewer 2 Report

After carefully reading this manuscript, I must say that, from my point of view, the authors have done research on an important and current topic: Different ways of activating the Transversus Abdominis muscle. This could be interesting to trainers and physical therapist injury prevention and to guide interventions.

This is an interesting aim in physical therapy and exercise research. I have considered the quality of the manuscript redaction and presentation, the quality of the research methodology and the novelty and importance of the observations.

Specific suggestions are provided below to assist in improving this manuscript.

Abstract.

After reading the abstract, and without reading the rest of the paper, I feel it needs somehow a clearer structure. In line 24, the conclusion is not easy to understand.

Introduction:

Line 39. Please, clarify the reference to Gorber et al. I don’t quite understand how low back pain patients can be classified as “asymptomatic”.

Lines 55,56. Please rephrase the sentences, the two affirmations seem somehow “pasted” together, making little sense.

Line 64-65. The final sentence would be better placed in Conclusions.

Materials and Methods.

Line 71. Please, I would like to know why women were excluded. Are there any significant gender differences to justify this choice?

Line 79. “Written explanation”.

Please, specify if the measure taken was unique or it was the mean of several measures taken.

Lines 109-111. The difficulty of the rating seems quite simplistic. Is there a previous published reference that supports this way of measuring difficulty?

Discussion

Lines 197 to 199. I do think this is a big methodogical mistake. Verbal instructions protocol should have been considered as a priority, the two verbal instructions are very different in terms of muscular action, and, in fact, the authors acknowledge variances in their results due to this verbal clues.

Line 230. In my opinion, If ADIM needs of visual feedback provided by ultrasound exploration to be properly learned, that implies too much technical and logistic conditions to make it a day-to-day technique for patients with low back pain or other pathologies that may benefit from trunk stabilization.

Lines 238-240. This part of the discussion may benefit of a explanation in plain language about the significance of the relation between thickness and subjective difficulty, s it’s difficult to understand the paragraph as it is written now.

Lines 244. I certainly don’t understand what do the authors mean by the effect of bed pressure.

Conclussions

Line 259. Please, analyze if the subjective difficulty of AB is related with the use of different verbal clues.

Line 266. I feel that if one of the conclusions is that posture may have more influence than the exercise type, this may overrule the rest of the conclusions of this study.

As a general conclusion, the paper needs a deep change in its structure in order to properly explain the results. As it is now, the reader feels confused about the conclusions, doubting if the results support one or none of them. A little English high quality editing may help with this, as some of the sentences seem “pasted” to each other.

Author Response

Dear Reviewer 2

We are very sorry for the late submission. I have attached a response file about the fix. Please see the attachment. Thank you very much.

Reviewer 3 Report

I think, that the Article will improve following information:

  1. why you realised this research?
  2. practical impact and possibilities of use of presented information

Author Response

Dear Reviewer 3 

We are very sorry for the late submission. I have attached a response file about the fix. Please see the attachment. Thank you very much.

Reviewer 4 Report

Dear Authors,

Thank you for the paper.

Please, some comments below:
Can you add an explanation of the examiner's qualifications?
Can you provide photos for the muscle measurements using the ultrasound device? It should be something like what Markus and his team presented before in this paper:
https://www.researchgate.net/publication/341176016_Automatic_Tracking_of_the_Muscle_Tendon_Junction_in_Healthy_and_Impaired_Subjects_using_Deep_Learning
Pictures of the measurement positions that were relied upon in the research in the lying and sitting positions and the method for determining the measurement points despite the different physical patterns of the participants.

In light of the limitations of the study, it is generally accepted.

Measurement sequences and procedures without pictures illustrating this make it difficult to imagine the accurate flow of the measurement process.

Thanks

Author Response

Dear Reviewer 4

We are very sorry for the late submission. I have attached a response file about the fix. Please see the attachment. Thank you very much.

Round 2

Reviewer 2 Report

The manuscript has been conveniently revised and improved.

Reviewer 3 Report

no comments - authors accepted all my suggestions